# Fabricating a PDA-Liposome Dual-Film Coated Hollow Mesoporous Silica Nanoplatform for Chemo-Photothermal Synergistic Antitumor Therapy

**DOI:** 10.3390/pharmaceutics15041128

**Published:** 2023-04-03

**Authors:** Chuanyong Fan, Xiyu Wang, Yuwen Wang, Ziyue Xi, Yuxin Wang, Shuang Zhu, Miao Wang, Lu Xu

**Affiliations:** 1School of Pharmacy, Shenyang Pharmaceutical University, Shenyang 110016, China; 2School of Pharmaceutical Engineering, Shenyang Pharmaceutical University, Shenyang 110016, China; 3School of Life Science and Biopharmaceutics, Shenyang Pharmaceutical University, Shenyang 110016, China

**Keywords:** chemo-photothermal therapy, hollow mesoporous silica, liposome-TPGS, polydopamine, doxorubicin

## Abstract

In this study, we synthesized hollow mesoporous silica nanoparticles (HMSNs) coated with polydopamine (PDA) and a D-α-tocopheryl polyethylene glycol 1000 succinate (TPGS)-modified hybrid lipid membrane (denoted as HMSNs-PDA@liposome-TPGS) to load doxorubicin (DOX), which achieved the integration of chemotherapy and photothermal therapy (PTT). Dynamic light scattering (DLS), transmission electron microscopy (TEM), N_2_ adsorption/desorption, Fourier transform infrared spectrometry (FT-IR), and small-angle X-ray scattering (SAXS) were used to show the successful fabrication of the nanocarrier. Simultaneously, in vitro drug release experiments showed the pH/NIR-laser-triggered DOX release profiles, which could enhance the synergistic therapeutic anticancer effect. Hemolysis tests, non-specific protein adsorption tests, and in vivo pharmacokinetics studies exhibited that the HMSNs-PDA@liposome-TPGS had a prolonged blood circulation time and greater hemocompatibility compared with HMSNs-PDA. Cellular uptake experiments demonstrated that HMSNs-PDA@liposome-TPGS had a high cellular uptake efficiency. In vitro and in vivo antitumor efficiency evaluations showed that the HMSNs-PDA@liposome-TPGS + NIR group had a desirable inhibitory activity on tumor growth. In conclusion, HMSNs-PDA@liposome-TPGS successfully achieved the synergistic combination of chemotherapy and photothermal therapy, and is expected to become one of the candidates for the combination of photothermal therapy and chemotherapy antitumor strategies.

## 1. Introduction

Breast cancer is still the leading cause of cancer death in women worldwide [1]. The traditional treatment methods (including chemotherapy, surgery, and radiotherapy) are still the main options for the treatment of breast cancer. However, the results of these therapeutic methods are dissatisfactory, due to the drug resistance caused by long-term chemotherapy, damage to normal tissues and organs caused by radiotherapy, and high recurrence rate after surgery. Simultaneously, due to the complex, diverse, and heterogeneous characteristics of tumors, single treatments (radiotherapy, chemotherapy, etc.) cannot achieve the ideal effect of tumor suppression. To overcome the limited therapeutic effect of single chemotherapy, the synergistic treatment of tumors by combining multiple antitumor strategies has attracted increasing attention [2,3,4,5,6,7,8]. More importantly, nanotechnology can enable multimodal synergistic therapies by assembling various therapeutic elements into a nanoplatform, thereby forming multifunctional nanomaterials [9,10,11]. In this regard, various synergistic nanoplatforms have been proposed, such as chemo-photothermal therapy (chemo-PTT) [12,13], chemo-photodynamic therapy (chemo-PDT) [14], chemoimmunotherapy [15,16], and PTT/PDT [17]. Among these, the combination of PTT and chemotherapy has attracted much attention due to its unique advantages in cancer treatment. Firstly, PTT is potentially specific and less invasive, because the PTT agent can effectively convert light energy into thermal energy to trigger destructive damage to tumor cells only when a certain body tissue is exposed to near-infrared (NIR) lasers [18,19,20,21]. Secondly, under NIR laser excitation, the local heat will improve the cell metabolism and cell membrane permeability, thereby promoting the uptake of anticancer drugs by cancer cells, effectively reducing the necessary dose of chemotherapeutic drugs to synergistically enhance the therapeutic effect and reduce drug resistance [22,23,24].

Therefore, how to construct an effective nanoplatform to achieve chemotherapy–photothermal (chemo-PTT) synergistic therapy is particularly important. On the one hand, an appropriate photothermal converter is required to induce mild hyperthermia to achieve chemo-PTT therapy for cancer. To induce the photothermal ablation of cancer cells, a significant number of NIR-absorbing inorganic nanomaterials, including black phosphorus [25], carbon nanomaterials (carbon nanotube, graphene oxide) [26], gold nanostructures [27], copper sulfide nanoparticles [28], Cu-based semiconductor nanoparticles [29], and two-dimensional antimonene [30], have been widely investigated. Nevertheless, these agents cannot apply to clinical applications stemming from the limited long-term safety [31,32]. Compared with the high photothermal conversion performance of inorganic photothermal agents, organic photothermal agents have attracted widespread attention due to their superior biocompatibility and biodegradability. In recent years, polydopamine (PDA)-based surface modification has shown many advantages including its easy adhesion to surfaces, simple preparation method, and excellent pH sensitivity, which make it an excellent gatekeeper [33,34]. Moreover, it also exhibited outstanding light-to-heat conversion efficiency under NIR irradiation [35]. On the other hand, various organic and inorganic nanomaterials, such as synthetic polymers (dendrimers) [36], carbon-based nanostructures (carbon nanotubes [37,38], graphene [39]), nanoscale metal-organic frameworks (ZIF-8) [40], and mesoporous silica (silica-based nanocapsules) [41] have been used to accomplish a synergistic chemo-PTT antitumor impact. Among these carriers, mesoporous silica NPs (MSNs) are commonly used to load antitumor drugs due to their stable porous structure, easy-to-adjust pore size, large specific surface area, high drug loading capability, and simplicity of surface functionalization [42]. Compared with MSNs, HMSNs with a hollow structure showed more obvious advantages in the field of drug delivery [43]. To date, different chemo-PTT systems based on HMSNs have been created [44,45]. However, designing an intelligent and manageable gatekeeper for the HMSNs to achieve accurate and effective drug delivery is still a significant difficulty.

Additionally, the application of silica-based nanoparticles as drug carriers has also faced many challenges, such as their easy aggregation in saline buffers, non-specific protein adsorption in the blood circulation, rapid clearance by the reticuloendothelial system (RES), and high rates of hemolysis [46,47,48]. Therefore, solving the above limitations of silica-based nanomaterials is particularly important to improve their antitumor effect. Inspired by the supported lipid bilayer that mimics the cell membrane, MSNs-supported lipid bilayers (protocells) nanocarriers show unique advantages in addressing the above-mentioned limitations of silica-based materials. Protocell structures can fully combine the advantages of liposomes and MSNs [49,50,51], as (1) the lipid layer can not only promote the internalization of cells but also improve the dispersion and biocompatibility of MSNs [52,53,54] and (2) MSNs can serve as a supporting skeleton to stabilize the lipid bilayer. In previous reports, Wang et al. designed a calcium carbonate (CaCO_3_) and lipid membrane (liposomes) double-coated mesoporous silica nanodrug delivery platform (MSNs@CaCO_3_ @liposomes) to enhance cellular uptake in terms of improving blood circulation efficiency and biocompatibility [55]. In addition, lipid bilayers modified with polyethylene glycol (PEG) can increase the time of circulation and decrease the immunogenicity of MSN, consequently boosting the aggregation of drug carriers at the tumor site via the enhanced permeability and retention (EPR) effect [56,57,58]. Therefore, we integrated D-α-tocopheryl polyethylene glycol 1000 succinate (TPGS) into the lipid layer to achieve the PEGylation of the lipid layer, thereby prolonging the blood circulation time of the carrier. TPGS, as a water-soluble derivative of natural vitamin E, has been widely used in tumor therapy in recent years. TPGS can act as an effective surfactant to emulsify hydrophobic molecules and stabilize nanoparticles. Furthermore, TPGS has been shown to improve the drug encapsulation efficiency, cellular uptake, and in vitro cytotoxicity of cancer cells and the reversal of multidrug resistance (MDR) [59]. More importantly, the PEG chain of TPGS can prolong the reaction time and systemic circulation time in the blood stream after intravenous injection, enabling the drug carrier to better accumulate in the tumor site via the EPR effect. Therefore, TPGS-modified mixed lipids have been widely used as drug carriers to improve the blood circulation time and promote drug enrichment at tumor sites. Song et al. formulated TPGS-modified long-circulating liposomes to load ZgI (ziyuglycoside I). Compared with blank liposomes, the ZgI-TPGS liposomes exhibited a significantly longer mean residence time (MRT) and significantly lower clearance (CL) rate [60].

Here, we constructed a hollow mesoporous silica nanodrug delivery system (DOX/HMSNs-PDA@liposome-TPGS) double-coated with polydopamine (PDA) and a hybrid lipid membrane (liposome-TPGS) for chemotherapy–photothermal synergistic therapy. As shown in Figure 1, we coated a PDA shell and hybrid lipid film on the surfaces of the HMSNs loaded with the anticancer drug doxorubicin (DOX), and the structure not only effectively prevented DOX leakage under physiological cycling conditions (pH 7.4), but also exhibited sustained-release behavior in the tumor microenvironment (pH 5.0). Meanwhile, the coating of liposome-TPGS can prolong the circulation time of the carrier and improve the blood compatibility of the carrier. Additionally, the PDA coating showed excellent photothermal conversion efficiency (η = 16.7%). Furthermore, the DOX/HMSNs-PDA@liposome-TPGS exhibited good chemo-PTT synergistic antitumor effects through in vitro and in vivo antitumor experiments. Our results all proved that the DOX/HMSNs-PDA@liposome-TPGS could be a promising nanoplatform for drug delivery and combination chemo-PTT therapy for cancer.

## 2. Materials and Methods

### 2.1. Materials

The cetyltrimethylammonium bromide (CTAB) and tetraethoxysilane (TEOS) were purchased from the Shandong YuWang Chemical Reagent Corporation (Yucheng, China). The cholesterol was obtained from Panjin Yanfeng Technology Co, Ltd. (Panjin, China). The D-α-tocopheryl polyethylene glycol 1000 succinate (TPGS) was bought from Kunshan Rongbai Biotechnology Co, Ltd. (Kunshan, China). The F127 pluronic polymer, 4-(2-hydroxyethyl)-1-piperazineethanesulfonic acid (HEPES), 3-aminopropyltriethoxysilane (APTES), soybean lecithin (SPC), dopamine hydrochloride (DA), and doxorubicin hydrochloride (DOX) were purchased from Aladdin Chemistry Co., Ltd. (Shanghai, China). The 3-(4,5-dimethylthiazol-2-yl)-2,5-diphenyltetrazolium bromide (MTT) and 4% paraformaldehyde, 4′,6-diamidino-2-phenylindole (DAPI) were obtained from Dalian Meilun Biotechnology Company (Dalian, China).

### 2.2. Synthesis of HMSNs-PDA and HMSNs-PDA@Liposome-TPGS

The procedures for the HMSNs were conducted via a self-template etching method based on previous reports with a slight adjustment [61]. Here, 5 mL of F127 solution (5 mg/mL) and 7 mL of 2 M NaOH were added to a solution of 1 g of CTAB that had been dissolved in 475 mL of water. The reaction mixture was then stirred while being heated to 80 °C (600 rpm). Then, TEOS (6 mL) was rapidly added to this solution and the reaction mixture was stirred for 1 h. Next, TEOS (5 mL) was slowly added to this suspension in a dropwise manner. The reaction mixture was then agitated for a further hour. The above product was centrifuged, washed with water and absolute ethanol, and dried. The obtained powder was dissolved in PBS (pH 7.4) at a concentration of 0.5 mg/mL at 65 °C for 18 h under gentle stirring. The HMSNs were collected by centrifugation at 9000 rpm and washed with absolute ethanol three times.

Then, the amino-modified HMSNs (denoted as HMSNs-NH_2_) were fabricated using the after-grafting method. In brief, 0.2365 g of HMSNs was placed in 23.65 mL of ethanol and the mixture was sonicated to be dispersed. Then, APTES (710 μL) was added dropwise and the reactant was stirred for 24 h to synthesize the HMSNs-NH_2_. Afterwards, 10 mg of HMSNs-NH_2_ was placed in 5 mL of HEPES (pH 7.4). Next, 2 mg of dopamine and 2.4 mg of ammonium persulfate were added and reacted for 12 h. The product was collected and recorded as HMSNs-PDA.

The liposome-TPGS was synthesized using the thin-film hydration method [62]. Firstly, SPC, cholesterol, and TPGS (*w*/*w*/*w*, 8:1:1) were dissolved in 3 mL of chloroform and the organic solvent was removed by rotary evaporation under vacuum (−0.1 MPa) at 37 °C, after which a layer of thin film existed on the bottom. Then, the film was hydrated with deionized water and sonicated to obtain liposome-TPGS. Finally, HMSNs-PDA@liposome-TPGS was fabricated by mixing HMSNs-PDA and liposome-TPGS (*w*/*w*, 1:2), followed by sonication for 1 min.

### 2.3. Characterization of NPs

#### 2.3.1. Transmission Electron Microscopy (TEM)

We prepared the sample into a 2 mg/mL ethanol solution, then added the prepared sample dropwise on the copper grid coated with carbon and let it stand for 1 min. Then, it was dried under an infrared lamp before being photographed. For HMSNs-PDA@liposome-TPGS samples coated with liposome-TPGS mixed lipid film, we first let the sample sit for 1 min. Then, the grid was incubated with 2% phosphotungstic acid for 1 min and dried under infrared light to shoot. The morphology was observed by TEM (JSM-6510A, JEOL, Tokyo, Japan) at an acceleration voltage of 200 kV.

#### 2.3.2. N_2_ Adsorption/Desorption

Approximately 0.1 g of finely ground sample was placed in a sample tube. The sample was then pretreated at the appropriate temperature. After the pretreatment, the sample was weighed again to obtain the exact mass of the sample. Finally, a nitrogen adsorption/desorption isotherm test was performed. The nitrogen adsorption and desorption capacity was measured on an SA3100 surface with a pore size analyzer (Beckman Coulter, Brea, CA, USA).

#### 2.3.3. Fourier Transform Infrared Spectroscopy (FT-IR)

We took appropriate amounts of the samples to be tested. The samples were prepared using the KBr pellet method. Fourier transform infrared spectroscopy (FT-IR) was performed using a spectrometer (Spectrum 1000, PerkinElmer, Waltham, MA, USA). For the test conditions, the wavelength range was 4000~400 cm^−1^ and the resolution was 4 cm^−1^.

#### 2.3.4. Small-Angle X-ray Scattering (SAXS)

The powder was placed in the cuvette and measured after the instrument had reached vacuum conditions. The test conditions were a scanning range of 0°~6° (2θ), scanning step size of 0.02°, and scanning speed of 0.6°/min. An SAXS study was carried out to investigate the state of the drug (crystalline/amorphous) in the HMSNs-PDA.

#### 2.3.5. Size Distribution and Zeta Potential (ζ)

We diluted the samples and put them into the sample cell. The size and charge of the NPs were measured using a Zetasizer Nano ZS90 instrument (Malvern Instruments Ltd., Malvern, UK).

### 2.4. Photothermal Conversion Property Test

To determine the photothermal effects of different samples, water, HMSNs-NH_2_, HMSNs-PDA, and HMSNs-PDA@liposome-TPGS at different concentrations (25, 50, 100, 200, and 400 µg/mL) were subjected to NIR irradiation at 2.0 W/cm^2^ for 5 min (808 nm) and the HMSNs-PDA@liposome-TPGS (400 µg/mL) sample suspensions were irradiated with various power densities for 5 min. To investigate the photothermal stability of HMSNs-PDA@liposome-TPGS, the sample suspensions were continuously irradiated with an NIR laser for 5 min (808 nm, 2 W/cm^2^) and then cooled naturally for 10 min without irradiation. The temperature changes of the HMSNs-PDA@liposome-TPGS with three on/off cycles of laser irradiation were recorded. Additionally, the photothermal conversion efficiency of the HMSNs-PDA@liposome-TPGS was calculated by referring to the relevant literature [63,64,65].

### 2.5. Drug Loading Capacity and Encapsulation Efficiency

Here, 10 mg of HMSNs-PDA and 5 mg of DOX were dispersed in 3 mL of phosphate-buffered solution (PBS, pH 7.4) and stirred for 24 h in the dark, then the precipitate was collected and washed three times with PBS. The washed PBS was collected and a UV spectrophotometer was used to determine the concentration of DOX in the PBS at 480 nm. The precipitates were dried to obtain drug-loaded HMSNs-PDA (denoted as DOX/HMSNs-PDA). To synthesize the drug-loaded HMSNs-PDA@liposome-TPGS, the DOX/HMSNs-PDA was coated with a lipid membrane as described above, and the HMSNs-PDA was replaced with DOX/HMSNs-PDA. Meanwhile, all other operations remained unchanged. The formula of the drug loading (*DL%*) was as follows:(1)$$DL\left(\%\right)=M_{drug\;in\;NPs}/\left(M_{total\;drug}+M_{NPs}\right)\times 100\%$$
where *M_drug in NPs_* is the mass of DOX-loaded in nanoparticles, *M_total drug_* is the initial mass of DOX, and *M_NPs_* is the dry weight of the different nanoparticles.

### 2.6. In Vitro Drug Release Experiment

The dialysis method was conducted to study the release profiles of drug-loaded NPs. The DOX/HMSNs-PDA@liposome-TPGS (0.5 mg/mL) suspension was added to dialysis bags (MWCO = 10,000 Da) and placed in separate flasks containing 30 mL of PBS solution at different pH levels (pH 7.4, pH 5.0). The flasks were then shaken in a gas bath shaker (37 °C, 100 rpm). At predetermined time points, 3 mL of release medium was withdrawn and an equal amount of corresponding blank release medium was supplemented. Meanwhile, the irradiation groups were exposed to NIR (808 nm, 2 W/cm^2^) for 5 min each hour. Finally, the amount of released DOX was determined at 480 nm using ultraviolet spectroscopy (UV-756PC, Sunny Hengping Instrument Co., Ltd., Shanghai, China).

### 2.7. MTT Assay

The 4T1 cells were cultured into 96-well plates at a density of 1×10^4^ cells/well. Different concentrations of blank carriers (HMSNs-NH_2_, HMSNs-PDA, and HMSNs-PDA@liposome-TPGS) and DOX-loaded carriers (DOX/HMSNs-NH_2_, DOX/HMSNs-PDA, and DOX/HMSNs-PDA@liposome-TPGS) were co-incubated with cells for 24 h. The DOX/HMSNs-PDA@liposome-TPGS + NIR group was incubated for 4 h and irradiated for 5 min (808 nm, 2 W/cm^2^), then cultured for another 20 h. The culture medium was removed and washed with PBS after incubation. Then, 100 µL of MTT solution (0.5 mg/mL) was added to each well and incubated for an additional 4 h. Finally, the MTT solution was discarded and 200 μL of DMSO solution was added to each well and shaken for 10 min at 37 °C to fully dissolve the formazan crystals. The absorbance was measured at 490 nm using a microplate reader.

### 2.8. Blood Compatibility Experiment

#### 2.8.1. Hemolysis Test

Hemolysis caused by NPs was evaluated using a hemolysis test. Samples (HMSNs-NH_2_, HMSNs-PDA, HMSNs-PDA@liposome-TPGS were dispersed in saline) containing a 0~800 μg/mL concentration gradient were incubated with 2% erythrocyte suspensions in an equal volume for 3 h at 37 °C. The 2% erythrocyte suspensions mixed with saline or deionized water were regarded as negative control or positive control groups, respectively. The mixture was centrifuged at 2000 rpm to collect the supernatant. The absorbance of the supernatant was measured at 540 nm and the hemolysis ratio was calculated by the following equation:(2)$$\mathrm{Hemolysis}\;\mathrm{ratio}\;\left(\%\right)=\left(A_{S}-A_{N}\right)/\left(A_{P}-A_{N}\right)\times 100\%$$
where *A_S_* is the absorbance of each sample, *A_N_* is the absorbance of the negative control, and *A_P_* is the absorbance of the positive control.

#### 2.8.2. Non-Specific Protein Adsorption Test

Bovine serum albumin (BSA) was chosen as a model protein to evaluate the adsorption of NPs. HMSNs-NH_2_, HMSNs-PDA, and HMSNs-PDA@liposome-TPGS were incubated with BSA solution (PBS, 0.5 mg/mL) for 6 h at 37 °C, respectively. After incubation, the mixture was centrifuged to obtain the supernatant. Then, 200 µL of supernatant was added to 2 mL of Coomassie brilliant blue dye, shaken for 30 s, and placed at room temperature for 3 min. The absorbance of the solution was measured at 595 nm, and the adsorption rate Q (%) was calculated using the following equation:(3)$$Q\;\left(\%\right)=\left(C_{0}-C\right)\times V/m\times 100\%$$
where *C*_0_ and *C* are the initial and remaining concentrations of the BSA solution, *V* is the volume of the solution, and m is the quality of the nanoparticle sample. Each experiment was carried out in triplicate.

### 2.9. Cellular Uptake Evaluation

The 4T1 cells at the logarithmic growth stage were cultured into 12-well plates at a density of 1×10^5^ cell/well and incubated overnight in an incubator with 5% CO_2_ at 37 °C for adherence. DOX, DOX/HMSNs-PDA, DOX/HMSNs-PDA@liposome-TPGS, and DOX/HMSNs-PDA@liposome-TPGS + NIR were incubated with cells for 4 h. For the DOX/HMSNs-PDA@liposome-TPGS + NIR group, after 2 h of incubation, the sample was irradiated with an 808 nm laser at a power density of 2 W/cm^2^ for 5 min and cultured for another 2 h. The medium was removed and washed with PBS. Next, the cells were fixed with 4% paraformaldehyde for 15 min and stained with 200 μL DAPI for 10 min. Eventually, the samples were washed with PBS and observed under a confocal laser scanning microscope (CLSM).

### 2.10. In Vivo Antitumor Effect Study and H&E Staining Analysis

The tumor-bearing BALB/c mice were randomly divided into five groups, including a saline group, DOX group, DOX/HMSNs-PDA group, DOX/HMSNs-PDA@liposome-TPGS group, and DOX/HMSNs-PDA@liposome-TPGS + NIR group (n = 5). Sample solutions at a 5.0 mg/kg equivalent dose of DOX were intravenously administrated and the mice were weighed every other day. For the DOX/HMSNs-PDA@liposome-TPGS + NIR group, the group was irradiated for 5 min (808 nm, 2 W/cm^2^) after 6 h of administration. At the end of the experiment, the mice were sacrificed and the heart, liver, spleen, lung, kidney, and tumor tissues were collected, weighed, and fixed with a 4% paraformaldehyde solution. After dehydration, paraffin embedding, and sectioning, the samples were stained with hematoxylin and eosin (H&E). The pathological changes of various tissues and organs were observed under an optical microscope.

### 2.11. Biodistribution Behavior In Vivo and Internal Long-Circulation Performance

To investigate the distribution of the nanoformulations in mice, ICG was loaded into the nanoplatform and stirred in the dark for 24 h, then the product was collected by centrifugation. When the tumor volume reached about 200 mm^3^, 100 μL of ICG/HMSNs-PDA and ICG/HMSNs-PDA@liposome-TPGS (administered dose 50 mg/kg) were injected via the tail vein. The fluorescence imaging system (IVIS Lumina) was used to perform PA imaging at 3 h and 24 h. Then, the mice were sacrificed to observe the fluorescence of the heart, liver, spleen, lung, kidney, and tumor samples (excitation/emission wavelength 720 nm/790 nm).

In order to reflect the long-cycle capacity of the carrier, SD rats were injected intravenously with ICG-labeled nanocarriers. The SD rats were anesthetized and blood samples were collected from the orbit at predetermined time points (5 min, 30 min, 1 h, 2 h, 4 h, 6 h, 8 h, 12 h, 24 h, and 48 h). The blood samples were centrifuged for 10 min at 4000 rpm and the supernatant was taken for fluorescence detection using a microplate reader. To quantify the relative percentage of remaining NPs in the blood circulation after injection at each blood sample time point, the relative fluorescence signal (the ratio of the fluorescence intensity of the blood sample at each time point to the fluorescence intensity of the blood sample at 5 min) was used to represent the systemic capacity of the carrier.

### 2.12. Statistical Analysis

The experiments were conducted at least in triplicate. Statistical differences between the two groups were analyzed by *t*-test. When comparing multiple groups of data, statistical differences between groups were tested by one-way ANOVA. The statistical significance levels were set at * *p* < 0.05, ** *p* < 0.01, *** *p* < 0.001, and **** *p* < 0.0001.

## 3. Results and Discussion

### 3.1. Synthesis and Characterization of NPs

The HMSNs-PDA and HMSNs-PDA@liposome-TPGS were synthesized as described in Section 2.2. The TEM images (Figure 1A–D) and particle size distribution curves (Figure 2A–C) of different samples were shown here. The TEM images indicated that the bare HMSNs displayed a spherical shape and hollow mesoporous structure. The hydrated particle size of the HMSNs was about 122.5 ± 14.93 nm (PDI 0.221) (Table 1). The outer shell was about 7 nm thickness. After coating with PDA, the surface of the HMSNs-PDA became rough and mesoporous channels were partially masked by a thin PDA film. After further coating with a lipid film, a lipid layer with a thickness of about 10 nm appeared on the surface of the HMSNs-PDA@liposome-TPGS. Meanwhile, the particle sizes of HMSNs-PDA and HMSNs-PDA@liposome-TPGS increased to 153.8 ± 25.18 nm (PDI 0.128) and 220 ± 16.3 nm (PDI 0.216), respectively. Since the TPGS embedded in the lipid membrane had a PEG hydrophilic end, it could form a hydration layer on the surface of the HMSNs-PDA, thereby making it difficult for the particles to aggregate. Therefore, the HMSNs-PDA@liposome-TPGS exhibited better dispersion. Similarly, the average surface charges of the different samples also changed significantly (Figure 2D). The surface charge of the bare HMSNs was (−18.13 ± 3.17 mV), due to the large number of silane alcohols on silica surface. The surface charge of HMSNs-NH_2_ then changed from negative to positive after the surface amination step. The HMSNs-NH_2_ showed a positive potential of 14.23 ± 3.01 mV. When the HMSNs-NH_2_ was coated with PDA, the potential of HMSNs-PDA decreased to −8.99 ± 0.02 mV, due to the presence of hydroxyl groups of PDA. After being coated with liposome-TPGS, the surface ζ potential of HMSNs-PDA@liposome-TPGS (−13.22 ± 3.09 mV) was close to the lipid membrane potential (−14.75 ± 1.06 mV). These results indicated that the PDA layer and liposome-TPGS hybrid lipid film were successfully wrapped on the surface of the HMSNs.

In addition, the N_2_ adsorption–desorption isotherms with the corresponding BJH pore size distribution were examined and were shown in Figure 2E,F. The specific surface area, pore volume, and most probable pore size values of different HSMNs samples were also shown in Table 2. The HMSNs displayed a type IV isotherm with a H2 hysteresis loop, which exhibited an obvious mesoporous feature. After coating with PDA, the specific surface area, pore volume, and pore diameter of HMSNs-PDA were significantly decreased due to the blocking of the PDA layer. After loading the drugs, the relevant parameters of DOX/HMSNs-PDA further decreased. These results confirmed that DOX occupied the mesoporous channels and the surface was modified by the PDA film successfully. Next, the surface characterization of prepared nanoparticles was also evaluated based on the FT-IR spectra (Figure 2G). The typical FT-IR absorption peaks of silica were found at 3422.0 cm^−1^, 1090.6 cm^−1^, and 466.2 cm^−1^, which were attributed to the stretching vibration of the Si-OH and the bending stretching vibration of the Si-O-Si, respectively. For the HMSNs-NH_2_, the FT-IR absorption showed N-H scissoring at 1628.2 cm^−1^, which was the characteristic absorption of the amino group. After further modification of the PDA, the surface of HMSNs-PDA showed a bending vibration peak of phenolic O-H at 1384.1 cm^−1^. The above indicated the synthesis of HMSN framework and the successful wrapping of the PDA layer.

Moreover, the pore orders of the HMSNs, HMSNs-NH_2_, and HMSNs-PDA were characterized by SAXS. In Figure 2H, it can be seen that the SAXS curves of the three carriers do not show the maximum peak value, which suggests that the channels of the three carriers were disordered. An XRD study was carried out to investigate the state of the drug (crystalline/amorphous) in HMSNs-PDA (Figure 2I). The DOX and physical mixture groups showed sharp and intense crystalline diffraction peaks at 2θ values of 12.94°, 14.80°, 15.56°, 18.40°, 22.46°, and 25.02°. Meanwhile, no crystal diffraction peaks were observed in the pattern of HMSNs-PDA. Surprisingly, the crystalline diffraction peaks of DOX were also not observed in the pattern of DOX/HMSNs-PDA. This phenomenon showed that the DOX was altered from a crystalline to an amorphous form during the preparation process. This was due to the adsorption of the drug into the pores, and the nanoscale pore size limited the long-range ordered structure associated with the existence of drug crystallization, thereby inhibiting the crystallization of the drug and making the drug exist in an amorphous state [66].

### 3.2. Photothermal Conversion Property Test of NPs

As shown in Figure 3A, the temperature gradient (ΔT) increased gradually to 6.5, 7.3, 7.7, 10.3, and 14.1 °C under 5 min of irradiation (808 nm, 2.0 W/cm^2^) with the increased NPs concentrations from 25 to 400 μg/mL. Meanwhile, the carrier exhibited a power-dependent temperature increase (Figure 3C). Moreover, the photothermal curves of water and different carriers with NIR irradiation (808 nm, 2.0 W/cm^2^) were exhibited in Figure 3B. After 5 min of irradiation at a power density of 2.0 W/cm^2^, the temperature gradient (ΔT) values observed in the HMSNs-PDA@liposome-TPGS and HMSNs-PDA were 13.8 °C and 13.9 °C, respectively. Conversely, no significant temperature fluctuations were detected in the HMSNs-NH_2_ suspension and water. This showed that the PDA-coated HMSNs had a good photothermal effect, and the coating of the liposome-TPGS did not affect the photothermal conversion performance of the HMSNs-PDA. To assess the photostability of the HMSNs-PDA@liposome-TPGS nanocarrier, we tracked the temperature changes over three cycles of laser on/off operations. As shown in Figure 3D, we observed no significant decreases in temperature fluctuations during repeated photothermal heating (300 s)–natural cooling cycles. This indicated that the HMSNs-PDA@liposome-TPGS nanocarrier was stable throughout the photothermal conversion process. In addition, based on the obtained data (Figure 3E,F), we calculated the photothermal conversion efficiency (η) to be 16.7%.

### 3.3. Drug Loading and In Vitro Release

The DL values of DOX/HMSNs-NH_2_, DOX/HMSNs-PDA, and DOX/HMSNs-PDA@liposome-TPGS were 29.96%, 32.20%, and 27.83%, respectively. The relatively low DOX release levels from DOX/HMSNs-NH_2_ (28.92%), DOX/HMSNs-PDA (20.74%), and DOX/HMSNs-PDA@liposome-TPGS (17.45%) were observed under physiological conditions (pH 7.4) (Figure 3G). However, in acidic conditions (pH 5.0), to simulate the endo-lysosomal environment of the cancer cells, the release of DOX in the three carriers increased significantly (90.84%, 83.36%, and 71.64%, respectively) (Figure 3H). This pH-responsive drug release behavior was caused by two factors. On the one hand, the solubility of DOX increased with decreasing pH [67]. On the other hand, the interaction between the DOX and mesoporous silica became weaker under acidic conditions, which also provided more favorable diffusion conditions for the DOX [68]. In addition, compared with DOX/HMSNs-NH_2_ and DOX/HMSNs-PDA, the drug release rate of DOX/HMSNs-PDA@liposome-TPGS decreased slightly under two different pH conditions. The cumulative release rates of DOX after 48 h were 21.35% and 69.82% at pH 7.4 and 5.0, respectively. This highlighted that the HSMNs coated with liposome-TPGS showed a slower drug release rate compared to the bare HMSNs. The sustained release could be attributed to the PDA and liposome film coating of the NPs, which prevented the premature release of DOX from the inner pores. After the NIR laser irradiation, the cumulative release amount of DOX reached 82% in acidic conditions (pH 5.0), which was higher than the 70% rate without laser irradiation at pH 5.0, and also higher than the 33.5% upon NIR laser irradiation in neutral conditions (Figure 3I). This property of NIR-responsive drug release was mainly attributed to the fact that the heat generated by the PDA under NIR laser irradiation destroyed the interaction between the DOX and HMSNs. Therefore, the HMSNs coated with PDA and hybrid lipid membrane can not only reduce the release of DOX in normal cells but also endow the nanoplatform with pH/NIR-responsive drug delivery capability, thereby achieving specific drug release at tumor sites.

### 3.4. Cell Toxicity Test

An MTT assay was carried out to evaluate the cell viability and study the chemo-photothermal therapeutic effect of DOX/HMSNs-PDA@liposome-TPGS. As can be seen from Figure 4A, the HMSNs-NH_2_ treatment group showed obvious cytotoxicity at a high concentration (120 μg/mL), and the cell survival rate was only 68.28%. However, the survival rate of the 4T1 cells remained above 80% after incubation with HMSNs-PDA or HMSNs-PDA@liposome-TPGS for 24 h at a concentration range of 15–120 μg/mL, which clearly demonstrated the negligible toxicity against tissues and cells. This showed that the HSMNs double-coated by PDA and a lipid membrane had good biocompatibility. Upon drug loading, the 4T1 cells were incubated with various concentrations of NPs at a series of DOX concentrations. As shown in Figure 4B, all formulation groups exhibited concentration-dependent killing effects. Since the free DOX group existed in the form of solution, it could quickly cross the cell membrane and enter the nucleus to play its role, so the cell survival rate was only 19.78% when the dosage was 20 µg/mL. However, there was a time delay in the release of DOX from HMSNs-NH_2_ and HMSNs-PDA groups, so the cell viability levels of both groups (45.54%, 63.07%) were higher than the free DOX group, and the cell survival rate of the DOX/HMSNs-PDA group was further increased because the pores were covered by the polydopamine coating. Nevertheless, the cell viability of the DOX/HMSNs-PDA@liposome-TPGS decreased to 39.45%, which may have been related to the mitochondrial pathway of TPGS regulating apoptosis [69,70]. It is worth noting that the cytotoxicity of the DOX/HMSNs-PDA@liposome-TPGS increased dramatically, with a cell viability level of 24.30% at a DOX concentration of 20 µg/mL after NIR laser irradiation. The above results could be quantitatively illustrated in terms of the IC_50_ values by using the GraphPad Prism software, and these IC_50_ values are listed in Table 3. As anticipated, the IC_50_ value of the DOX/HMSNs-PDA@liposome-TPGS decreased rapidly (by approximately 40%) with NIR irradiation compared to the group without it. This outcome can be attributed to the outstanding photothermal effect and enhanced drug release triggered by mild hyperthermia. In conclusion, the DOX/HMSNs-PDA@liposome-TPGS has good tumor cell growth inhibition ability under the assistance of near-infrared light.

### 3.5. Hemocompatibility Analysis

#### 3.5.1. Hemolysis Test

The hemocompatibility of nanoparticles must be considered for intravenous drug delivery systems. Therefore, we evaluated the hemocompatibility of HMSNs-PDA@liposome-TPGS using rat RBCs. As shown in Figure 4C,D, the hemolytic activity of three NPs were displayed in a dose-dependent manner within the concentration range (50–800 μg/mL). Among them, the hemolysis of the HMSNs-NH_2_ was the most significant, reaching 16.14 ± 2.65% at a concentration of 200 μg/mL. Meanwhile, complete hemolysis was observed once the concentration of HMSNs-NH_2_ exceeded 400 μg/mL. As reported by other studies, the hemolysis activity of bare HMSNs-NH_2_ was attributed to the interaction of the surface residual silanol (Si-OH) with quaternary ammonium ions in the erythrocyte membrane [71]. After wrapping the PDA and liposome-TPGS layers, the HMSNs-PDA@liposome-TPGS-treated samples did not cause any visible hemolysis at any tested concentration (50–800 μg/mL). The hemolysis rate was only 10% at concentrations up to 800 µg/mL. Therefore, the enhanced hemocompatibility of the HMSNs-PDA@liposome-TPGS resulted from the shielding of silanol by the PDA and liposome-TPGS layers.

#### 3.5.2. Non-Specific Protein Adsorption Test

When carrier materials enter the body, the non-specific proteins will be adsorbed on the surface of the material, resulting in the failure of the nanocarrier function [72,73]. Therefore, carrier materials should have a low protein adsorption rate to ensure the reliability of the nanocarrier transport in vivo. Here, we chose bovine serum albumin (BSA) as a model protein to investigate the non-specific protein adsorption of HMSNs-NH_2_, HMSNs-PDA, and HMSNs-PDA@liposome-TPGS, quantified by the BSA standard curves. The results are shown in Figure 4E. The protein adsorption capacity levels of HMSNs-NH_2_, HMSNs-PDA, and HMSNs-PDA@liposome-TPGS were 180.5 µg/mg, 208.0 µg/mg, and 146.0 µg/mg, respectively. The strong protein adsorption levels of the HMSNs-NH_2_ and HMSNs-PDA were attributed to the hydrogen bonds formed between the amino groups on the mesoporous silica surface and proteins, and the covalent bonding between the benzene rings contained in the PDA layer and aromatic amino acids, respectively. However, the addition of a lipid film shielded the influence of the amino group and PDA layer, resulting in a decrease in the adsorption capacity. Therefore, the coating of liposome-TPGS is beneficial to improve the biocompatibility of the carrier and reduce the possibility of its clearance by immune cells.

### 3.6. Cellular Uptake Evaluation

To observe the uptake of free DOX and DOX/HMSNs-PDA@liposome-TPGS nanoparticles by cells, confocal laser scanning microscopy (CLSM) was utilized. The images in the first row of Figure 4F show that after 4 h of incubation, the red fluorescence of the free DOX was primarily found in the nucleus. This could be attributed to the fact that free small DOX molecules are able to easily penetrate biological membranes through passive diffusion. However, endocytosis is generally considered one of the main entry mechanisms for various drug nanocarriers, which is slower than diffusion, so the DOX/HMSNs-NH_2_ and DOX/HMSNs-PDA treatment groups only showed weak red fluorescence. In comparison, the red fluorescence of the DOX/HMSNs-PDA@liposome-TPGS treatment group was significantly enhanced. This was because the lipid membrane had good fluidity and the affinity between the HMSNs and cell membrane increased after being encapsulated by a lipid membrane, resulting in enhanced uptake into the cell [74]. After exposure to the NIR laser, there was a notable increase in DOX fluorescence compared to the group that was not exposed to NIR illumination. It was shown that the temperature elevation has a positive impact on the drug release when near-infrared light irradiation is utilized, which is consistent with the results for in vitro release. In addition, it was also observed that the co-localization of DOX and the nucleus is more significant. This may be attributed to the increase in the permeability of the cell membrane with the increase in temperature, providing a superior condition for NPs to penetrate the cell membrane [75]. Taken together, this indicated that DOX/HMSNs-PDA@liposome-TPGS could be internalized and NIR could accelerate the drug release by enhancing the cell membrane penetrability and sensitivity, synergistically exerting photothermal–chemotherapy effects to kill tumor cells.

The average fluorescence intensity of DOX in each group shown by the Image Fiji analysis (Figure 4G). The DOX uptake of the laser irradiation group was 2.60 times that of the non-irradiated group, indicating that light irradiation can promote the intracellular drug release of the carrier. Pearson’s coefficient (Pearson’s R) is widely used to analyze the degree of correlation between two variables, and the closer the value is to 1, the higher the positive correlation between the two. Therefore, the colocalization of the nucleus and DOX was evaluated using the Pearson coefficient as an index. As shown in Figure 4H, the Pearson coefficient of DOX/HMSNs-PDA@liposome-TPGS + NIR group was 0.76, which was 1.52 times that of the non-irradiated group. This showed that light was favorable for DOX to enter the nucleus, so as to combine with deoxynucleotides in the nucleus to exert its medicinal effect.

### 3.7. In Vivo Antitumor Effect Study and H&E Staining Analysis

The 4T1 cells were implanted into the subcutaneous tissue of BABL/c mice. After tumor formation, the mice were peritumorally injected with solutions of saline, DOX, DOX/HMSNs-PDA, and DOX/HMSNs-PDA@liposome-TPGS, respectively. As shown in Figure 5A,B, each mouse group’s body weight did not significantly change throughout the course of the two-week experiment. On the contrary, there was a significant difference in tumor volume among the experimental groups. The DOX/HMSNs-PDA@liposome-TPGS group had a better antitumor effect than the DOX/HMSNs-PDA group, which might because the encapsulation of the lipid membrane prolonged the in vivo circulation time and increased the uptake of the nanodrug delivery system. It is worth noting that the DOX/HMSNs-PDA@liposome-TPGS + NIR group exhibited the most superior antitumor ability due to the excellent light-to-heat conversion effect of PDA, and there was almost no change in tumor volume. At the end of the pharmacodynamic experiments, the 4T1 tumor-bearing mice were executed and the tumor tissues were removed, weighed, and photographed (Figure 5C,D). The results showed a consistent trend in the growth of the tumor mass and volume. The tumor mass of the DOX/HMSNs-PDA@liposome-TPGS + NIR group was about 1/5 that of the saline group. In addition, the tumor inhibition rate of the DOX/HMSNs-PDA@liposome-TPGS + NIR group reached 87.39%. This indicated that the DOX/HMSNs-PDA@liposome-TPGS achieved effective accumulation in the tumor site and a prominent tumor retention effect in the blood, showing an outstanding chemo-PTT therapeutic effect. Moreover, the H&E staining of tumor slices indicated that DOX/HMSNs-PDA@liposome-TPGS + NIR caused more extensive apoptosis and necrosis than other treatments. Meanwhile, no discernible histological damage was found when the major organs were H&E-stained (Figure 5E). The above results confirmed the superior biocompatibility of the nanoplatform.

### 3.8. Biodistribution Behavior In Vivo and Internal Long-Circulation Performance

As shown in Figure 5F, the relative fluorescence signal of the ICG/HMSNs-PDA@liposome-TPGS was 48.7% at 24 h post-injection. However, only 31.4% of the ICG/HMSNs-PDA was retained in the blood circulation. The relative fluorescence intensity of ICG/HMSNs-PDA@liposome-TPGS group was 1.55 times that of the ICG/HMSNs-PDA group. Meanwhile, to investigate the distribution of the preparation in mice, the mice were sacrificed at 3 h and 24 h after the administration of ICG labeling carriers, and heart, liver, spleen, lung, kidney, and tumor tissues were collected for in vivo tissue imaging (Figure 5G). Compared with ICG/HMSNs-PDA, ICG/HMSNs-PDA@liposome-TPGS was more enriched in the tumor site, and a large number of carriers still existed in the tumor site after 24 h. Furthermore, the fluorescence intensity of ICG/HMSNs-PDA@liposome-TPGS group was 1.73 times that of the ICG/HMSNs-PDA group. These results showed that the PEG hydrophilic chain in TPGS can effectively decrease plasma protein adsorption and prolong the circulation time of nanoparticles in vivo, thereby enriching the tumor sites.

## 4. Conclusions

We developed a multifunctional nanoplatform, DOX/HMSNs-PDA@liposome-TPGS, for the controlled and precise delivery of the antitumor drug DOX, thereby achieving a chemo–photothermal synergistic antitumor effect. In this design, the HMSNs exhibited high drug loading (27.83%) due to their uniform mesoporous pore and cavity structures. After loading the DOX, the DOX/HMSNs-PDA@liposome-TPGS exhibited the expected pH/NIR-responsive drug release performance. Then, a PDA shell and hybrid lipid membrane were used as gatekeepers to cap the surface of the nanoparticles to achieve excellent photothermal properties and prolong the blood circulation time. The in vitro and in vivo antitumor experiments showed that DOX/HMSNs-PDA@liposome-TPGS not only had good biocompatibility but also exhibited excellent tumor suppression under NIR irradiation through synergistic chemotherapy-PTT effects. In brief, the hollow mesoporous silica nanodrug delivery system with a dual coating of polydopamine and hybrid lipid membrane was prepared to not only fully combine the advantages of chemotherapy and PTT, but also to prolong the blood circulation time to promote the effective enrichment of chemotherapeutic drugs at the tumor site. Therefore, this is a promising nanoplatform for synergistic chemo-PTT antitumor effects.

## Data Availability

Not applicable.

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
