# Peer review of "Fabricating a PDA-Liposome Dual-Film Coated Hollow Mesoporous Silica Nanoplatform for Chemo-Photothermal Synergistic Antitumor Therapy"

_pharmaceutics, 2023, doi:10.3390/pharmaceutics15041128_

Round 1
Reviewer 1 Report
In the present article, authors demonstrated a PDA-liposome dual-film coated hollow mesoporous silica nanoplatform for chemo-photothermal synergistic antitumor therapy. Overall, the present manuscript is well-written and the authors successfully achieve their goal of study with the necessary experiments. The present strategy greatly overcomes the existing drug delivery issues on silica materials and helps to construct synergistic strategies on NPs. Therefore, I recommend it for publication after small minor issues.
1. Aothors stated, " Nevertheless, these agents cannot apply to clinical stem from the limited long-term safety". Cite references (https://doi.org/10.1039/D2BM00692H,https://doi.org/10.3390/jfb10010004, etc. )
2. Authors stated," Moreover, it also exhibited an outstanding light-to-heat conversion efficiency under NIR irradiation". cite references
Reviewer 2 Report
The manuscript entitled "Fabricating PDA-liposome dual-film coated hollow mesoporous silica nanoplatform for chemo-photothermal synergistic antitumor therapy" authored by Chuanyong Fan et al. reports a new platform of multifunctiional hollow drug-carriers for antitumoral therapy. The authors have developed lipid coated hollow mesoporous silica nanoparticles for sinergically chemo-photothermal treatment of tumors. The paper is interesting and well structured, presentation and anlaysis of data are consistent. I would suggest following changes\corrections\ improvements for final acceptance
1) A moderate English revision of text for removing few typos\imperfections and further improvement of figures quality presentation is recommended
2) A slight literature references update taking into account of competitors nanoparticles\nanocapsules\nanotubes for similar antineoplastic applications could enhance audiance interest and could strength quality of the manuscript
3) What was the rationale in the choice of type of cells and type of rats in vivo data? Are the carriers tested for specific tumors ? Please authors comment on these issues\points. Just for your note, unfortunately in the submission I cannot see Hemocompatibility data:t hey are missing in the pdf
Reviewer 3 Report
Xu et al. described the formulation and characterization of mesoporous silica nanoplatforms coated with PDA-liposomes and encapsulating DOX for chemo-photothermal synergistic antitumor therapy. Results described in this manuscript are of interest. However, the authors have to add in the “Materials paragraph” a full description of all the techniques and materials used (see comments below), and several results have to be deeper commented (see comments below). Moreover, all the abbreviations need to be defined as soon as they are used: for example, in the abstract, several abbreviations are not defined. Finally, the manuscript needs to be carefully read to correct all the typos, the English and the grammatical mistakes.
In view of these general comments and the ones given below, I do recommend the publication of this manuscript in Pharmaceutics after major revision.
Please find below, the specific remarks and/or questions that need to be addressed before any publication.
1. Page 1, Abstract: The abbreviations “TPGS”, “DOX” and “PTT” have to be defined.
2. Page 2, Introduction, line 32 “Chemotherapy … breast cancer.”: What about the surgery, radiotherapy … ? The authors have to shortly developed these points.
3. Page 2, Introduction, line 36: Which are the therapies combined for breast cancer treatments?
4. Page 2, Introduction, lines 37-40 “More importantly, … nanomaterials.”: The authors have to give at least one reference of a review article on the use of nanotechnology as multimodal drug delivery systems.
Moreover, the word “nanotechnology” is a common word and doesn’t need a capital letter at its beginning.
5. Page 3, Introduction, line52: I guess that “PTT” abbreviation has to be changed to “chem-PTT” one.
6. Page 3, Introduction, line 54 “synthetic polymers (dendrimers)”: The authors have to be careful when they are presenting synthetic polymers: they are indeed not always dendrimers! A dendrimer has a specific structure which is different from those of synthetic polymers.
7. Page 3, Introduction, lines 77-79: I think these two first sentences have too form only one, and that the point between “their properties” and “Such as easy…” has to be changed to a comma.
8. Page 3, Introduction, lines 82-83: Is the sentence “Therefore, … anti-tumor effect.” A question? If yes, and interrogation point is needed at its end.
9. Page 3, Introduction, lines 92-95: The authors have to add at least one reference on PEG (poly(ethylene glycol)) and the EPR effect first described by Maeda et al..
10. Page 3, Introduction, lines 51-94: These two parts have to be rearranged. Indeed, the authors start to present nanoplatforms (lines 51-63), then they moved to the presentation of various PTT agents (lines 63-76) and came back to nanoplatforms (lines 77-94). It is therefore complicated to follow.
11. Page 4, Introduction, line 121: I suppose that the authors first performed in vitro experiments. I therefore suggest that the authors write first “in vitro” followed by “in vivo”.
12. Page 5, Introduction, Scheme 1: The scheme 1 looks like a figure. Moreover, the authors have to modify the color of the background because it is difficult/impossible to see some drawing.
13. Page 5, Materials, line 135: At the beginning, the authors used the abbreviation “DOX” instead of “Dox”. They must therefore homogenize their abbreviations.
Moreover, in this part, the authors have to described in details the apparatus they used to characterize their nanoplatforms (type of apparatus, samples preparation, methods used for the measurements, etc.).
14. Page 5, Paragraph 2.2
- line 144: I suggest to the authors to change “aggressively” to “vigorously”.
- line 145: “600 rpm” is not a very high value for a very strong (aggressively) stirring.
- line 148: The product was washed with what?
- lines 154-156: This sentence needs to be reworded.
- line 158 “hydration method”: The authors have to add at least one reference describing this well-known method.
- line 160 “rotary evaporation”: I guess that it was under vacuum? If yes, it has to be precise and the vacuum value has to be given.
- lines 161 and 163 “sonication”: For sonication do the authors use a bath or a probe? What about the power used?
15. Page 5 Paragraph 2.3: how were prepared the TEM samples? What about the grid used? Were the samples stained?
16. Page 6, Paragraph 2.4, line 178: What was the laser wavelength?
17. Page 6, Paragraph 2.6, line 205: A dialysis membrane (or bag) is usually characterized by its molecular weight cut off (MWCO), not by a molar mass (MW)!
18. Page 7, Paragraph 2.7, line 217: What about the irradiation power and the laser wavelength?
19. Page 7, Paragraph 2.9
- line 252: The number “2” has to be in subscript.
- line 258: What does “CLMS” mean?
20. Page 8, Paragraph 2.11, line 283: “rpm” means “rounds per minute”. “/min” is therefore not necessary.
21. Pages 8-9, Paragraph 3.1: The data (diameters, PDI, zeta potential, etc.) have to be gathered in a table to be more easily readable. I therefore suggest to the authors to add such table.
22. Page 9, Paragraph 3.1, Figure 1: As mentioned above, samples’ preparation and analysis conditions must be described. Did the authors use a staining agent? If yes, which one?
23. Page 11, Paragraph 3.2, lines 366-368: This sentence needs to be reworded.
24. Page 12, Paragraph 3.3
- line 386 “No noticeable release of DOX …”: I do not agree. The drug release reached a plateau varying between 20 to 30% after almost 12 hours of incubation, and a burst effect can be observed within the first 12 hours of incubation.
- line 390: I agree and the burst effect is much more important and faster. The authors have to mention such burst effect and comment it. Why is a burst effect observed? How can this effect be minimized?
- lines 391-394: The authors have to add references describing such phenomena.
25. Page 13, Paragraph 3.4: Why was a cytotoxicity observed for HMSNs-NH2 nanoobjects? Is it already described in the literature?
26. Page 14, Paragraph 3.4, Table 2: How can the authors explain the higher cytotoxicity of DOX/HMSNs-PDA@liposomes-TPGS in comparison to DOX/HMSNs-PDA?
27. Page 14, Paragraph 3.5.1, lines 465-466: The first sentence of this paragraph needs to be reworded.
28. Page 15, Paragraph 3.5.2: I agree with the test with BSA, but what about the complement activation one which is usually studied?
29. Page 16, Paragraph 3.6, Figure 4: The quality of this figure must be improved in terms of readability of the legends in particular.
30. Page 17, Paragraph 3.7, line 555 “There was almost no change in tumor volume.”: For which conditions? I guess that it is for nanoplatform after NIR irradiation?
31. Page 19, Conclusions
- line 599: I suggest to the authors to be more moderated because it is obvious from drug release profile that a burst effect does exist.
- What about the long-term toxicity of remaining mesoporous silica NPs? Do the authors have any idea of their behavior in vivo over long-term?
